# Mediation Effect of Social Distancing on Neonatal Vitamin D Status and Related Clinical Outcomes during the Coronavirus Disease-19 Pandemic

**DOI:** 10.3390/nu16121858

**Published:** 2024-06-13

**Authors:** Jin Su Jun, Dong Joon Kim, Seung Chan Kim, Jung Sook Yeom, Ji Sook Park

**Affiliations:** 1Department of Pediatrics, College of Medicine, Gyeongsang National University, Jinju 52727, Republic of Korea; dldirlquf@hanmail.net (J.S.J.); kdj89182@hanmail.net (D.J.K.); jsyeom@gnu.ac.kr (J.S.Y.); 2Department of Pediatircs, Gyeongsang National University Hospital, Jinju 52727, Republic of Korea; 3Institute of Medical Science, Gyeongsang National University, Jinju 52727, Republic of Korea; 4Biostatics Cooperation Center, Gyeongsang National University Hospital, Jinju 52727, Republic of Korea; seungchan.statistics@gmail.com

**Keywords:** newborn, vitamin D deficiency, COVID-19, social distancing

## Abstract

Background: We analyzed the impact of social distancing (SD) on vitamin D status and associated morbidity in neonates during the coronavirus disease (COVID-19) pandemic. Methods: Serum levels of 25-hydroxy vitamin D (25OHD) and clinical characteristics of newborn infants before (2019) and during SD (2021) were compared. Results: A total of 526 neonates (263 in 2019 and 263 in 2021) were included. The rate of vitamin D deficiency in neonates (47.1% vs. 35.4 %, *p* = 0.008) decreased and the rate of maternal vitamin D intake increased (6.8% vs. 37.6%, *p* < 0.001), respectively, during SD compared to those in 2019. The rates of hypocalcemia (12.5% vs. 3.8%, *p* < 0.001) and respiratory illness (57.0% vs. 43.0%, *p* = 0.002) decreased during SD. Neonatal vitamin D deficiency during SD was associated with maternal vitamin D supplementation (odds ratio [OR] = 0.463, *p* = 0.003) but was not associated with SD (OR = 0.772, *p* = 0.189). The mediation effect of SD on neonatal morbidity by neonatal vitamin D status was statistically insignificant. Conclusions: SD might affect the increased maternal vitamin D intake and decreased neonatal vitamin D deficiency. However, neonatal morbidity was not affected by SD, even with neonatal vitamin D status changes.

## 1. Introduction

Since the initial reports of the novel severe acute respiratory syndrome coronavirus 2 (SARS-CoV-2) in 2019 and the announcement of a pandemic by the World Health Organization (WHO) on March 11, 2020, people around the world have been affected by the coronavirus disease 2019 (COVID-19) [1]. Social distancing (SD) was applied worldwide as a nonpharmaceutical intervention to stop the transmission of SARS-CoV-2. It was also adopted in South Korea for over two years from March 2020 to April 2022, together with wearing a facial mask, and physical activity was significantly suppressed during SD [2]. Vaccines preventing the virus became available in February 2021, but they were recommended to pregnant women after eight months from their introduction in South Korea (www.kdca.go.kr; accessed on 21 October 2022). Because little was known about the vaccines, the vaccination rate of pregnant women was low at only 9.8% in Korea (National Health Insurance Service. Available at: http://www.ftoday.co.kr/news/articleView.html?idxno=231340; accessed on 1 April 2022). Although the vaccination rate was low, the infection rate was also lower in pregnant women (0.02%) than in others [3]. The lower infection rate, albeit low vaccination rate, might reflect the markedly decreased physical or social activity of pregnant women in Korea during SD. And decreased physical or social activity during SD resulted in an increased rate of vitamin D deficiency in Korean children and adolescents [4]. Therefore, SD also might affect vitamin D status in pregnant women. Neonatal vitamin D status can be affected by that of the mother during pregnancy, and neonates born from mothers with vitamin D deficiency are likely to show low serum vitamin D levels [5,6]. Vitamin D plays a fundamental role in promoting various physiological functions, including the immune system, and prenatal and postnatal lung growth, as well as calcium and phosphorus homeostasis in neonates [7,8]. Vitamin D deficiency in neonates is related to diverse morbidities, such as late-onset hypocalcemia, sepsis, and respiratory illness, including respiratory distress syndrome (RDS) and transient tachypnea of newborn (TTN) in both preterm and term newborn infants [9,10,11,12,13,14,15].

Vitamin D deficiency is a growing health concern in both adults and children, especially during the pandemic period. During SD, vitamin D deficiency and decreased levels vitamin D were reported in infants and children [16,17,18]. Low vitamin D levels increased the risk of severe COVID-19 progression and related hospitalization [19,20]. However, there is no report of change in vitamin D levels in neonates during the pandemic. Based on previous reports, we hypothesized that changes in maternal and neonatal vitamin D status during the period of SD might have occurred and that changes in neonatal vitamin D status might have induced changes in vitamin D-related neonatal morbidities compared to before SD. Therefore, we investigated neonatal vitamin D levels and clinical characteristics in 2019 (before SD, B-SD) and in 2021 (during SD, SD), respectively and statistically analyzed the effect of SD on the changes in vitamin D and related neonatal morbidities.

## 2. Materials and Methods

### 2.1. Study Population

The medical records of 874 neonates born and admitted to our hospital from January 2019 to December 2021 were retrospectively reviewed. Since SD was applied from 22 March 2020 to 18 April 2022, in South Korea, neonates born in 2020 were excluded in consideration of seasonal effects on vitamin D and social confusion at the beginning of SD. Since there was no statistically significant difference in preterm births with gestations of less than 32 weeks between 2019 and 2021 in this study (6.8% B-SD vs. 8.4% during SD, *p* = 0.622), we reviewed the clinical features of both preterm and term newborn infants. Gestational age (GA) and birth weight (BW), sex, Apgar score at 5 min, and delivery mode were reviewed as perinatal factors. Obstetric and neonatal factors were reviewed and defined as follows. Prolonged premature rupture of membrane (PPROM) was a rupture of the amniotic membrane over 18 h prior to giving birth that happened before 37 weeks of gestation; maternal hypertension (HTN) was pregnancy-induced hypertension, preeclampsia, eclampsia, or abruption of the placenta caused by hypertension; and maternal diabetes mellitus (DM) was gestational or underlying type 1 or 2 DM. Serum 25-hydroxy vitamin D (25OHD) levels, birth season, and prenatal maternal vitamin D intake were also investigated. Respiratory distress syndrome (RDS) was defined as diffuse ground glass opacity in both lung fields on chest X-ray with respiratory difficulties, and transient tachypnea of newborn (TTN) was a sunburst appearance in both hilar areas on chest X-ray with tachypnea [21]. Because both respiratory disorders can be associated with vitamin D deficiency in neonates and the rates of very preterm infants as a potent risk factor of RDS were not significantly different between the two periods, we considered RDS and TTN as a respiratory illness in this study [11,12,13,14]. Hypocalcemia was defined as total serum calcium <7.5 mg/dL in the first week of life. Sepsis was defined as a growth of pathogenic bacteria on blood culture and antibiotic treatment for over 5 days. NEC was defined as ≥stage 2 based on modified Bell’s criteria. BPD was defined as the need for oxygen or respiratory support at 36 weeks of gestation or the 28th day postnatally, according to the National Institute of Health Consensus severity-based definition for BPD. Neonatal jaundice was defined as unconjugated hyperbilirubinemia without underlying disorders such as hemolysis or genetic abnormality, for which phototherapy was indicated.

### 2.2. Vitamin D Measurement

We measured serum 25OHD concentrations by employing an electrochemiluminescence immunoassay method using an Elecsys and Cobas e 801 automated biochemical analyzer (Roche Diagnostics, Basel, Switzerland) when a newborn infant was admitted to our hospital. Vitamin D deficiency, insufficiency, and sufficiency were defined as a serum 25OHD concentration of <20, 20–30, and ≥30 ng/mL, respectively.

### 2.3. Statstical Analyses

Continuous variables were reported as the mean and standard deviation (SD). They were compared using an independent *t*-test, one-way analysis of variance (ANOVA), or the Mann–Whitney U test according to the number of analysis factors or the normality test results. The normality of the continuous data was determined using the Shapiro–Wilk test. Categorical variables were compared using the chi-square or Fisher’s exact test according to distribution assumption and reported as the number and percentage. To evaluate the factors associated with SD, multivariate linear or non-linear regression analyses were performed with significantly different factors found in univariate comparisons between the B-SD and SD groups after adjusting for BW. To investigate the factors associated with vitamin D deficiency in neonates, we conducted comparisons of the clinical characteristics according to neonatal vitamin D deficiency (serum 25OHD < 20 ng/mL). The process R package (version 0.2.6) running in Iavaan (version 0.6.12) was used to analyze the mediation effect of SD on clinical variables via neonatal vitamin D deficiency [22]. The data were analyzed using R software version 4.2.1 (R Core Team, R Foundation for Statistical Computing, Vienna, Austria, 2022). A two-sided *p*-value of <0.05 was considered significant. Dataset was uploaded as a Appendix A and can be downloaded.

## 3. Results

A total of 526 neonates whose serum vitamin D levels were tested within the first week after birth were included, which constituted 263 out of 326 (80.7%) admissions in 2019 and 263 out of 279 (94.3%) admissions in 2021. They were born at a mean of 36.5 ± 2.7 weeks of gestation and weighed 2670.3 ± 693.1 g, in total. One of them died due to pulmonary hypertension. The mean 25OHD of 526 neonates was 24.1 ± 11.9 ng/mL, which was measured on the day of 0.7 ± 2.0 after birth. The level of serum 25OHD was not correlated with GA (*r* = −0.029, *p* = 0.510). Serum 25OHD levels were higher during SD (2021) than B-SD (2019, 22.4 ± 11.0 vs. 25.8 ± 12.5 ng/mL, *p* = 0.001).

### 3.1. Associating Factors with Social Distancing in Neonates

Clinical characteristics were described and compared between the B-SD (2019) and SD (2021) groups (Table 1). GA and BW were not different between the two groups. Among the perinatal or obstetric factors, the rates of breastfeeding (BF, 14.8% vs. 36.4%, *p* < 0.001) and PPROM (16.7% vs. 24.7%, *p* = 0.031) decreased, but the proportions of male (67.7% vs. 54.4%, *p* = 0.002) and maternal vitamin D intake (37.6% vs. 6.8%, *p* < 0.001) increased during SD compared to B-SD. There was no seasonal difference at birth between the two periods. Among the neonatal factors, the rates of vitamin D deficiency (35.4% vs. 47.1%, *p* = 0.008), hypocalcemia (3.8% vs. 12.5%, *p* < 0.001), respiratory illness (43.0% vs. 57.0%, *p* = 0.002), and neonatal jaundice (6.1% vs. 11.8%, *p* = 0.033) were lower during SD than B-SD. Moderate-to-severe BPD increased in 2021 (2.3%) compared to 2019 (0.4%), but there was no statistical significance (*p* = 0.127). Multivariate regression analysis adjusted for BW was performed using significantly different clinical factors from univariate comparisons to analyze the factors associated with SD in neonates (Table 2). Maternal vitamin D intake (odds ratio [OR] = 8.735, *p* < 0.001), male sex (OR = 2.008, *p* = 0.001), PPROM (OR = 0.573, *p* = 0.028), and BF (OR = 0.383, *p* < 0.001) were associated with SD (Table 2). Neither neonatal vitamin D deficiency, hypocalcemia, nor neonatal jaundice was statistically associated with SD.

### 3.2. Associating Factors with Vitamin D Deficiency in Neonates

Clinical characteristics were compared according to neonatal vitamin D deficiency (25OHD < 20 ng/mL), insufficiency (20–30), and sufficiency (≥30), respectively (Table 3). The rates of vaginal delivery (VD, 33.2% vs. 27.8% or 19.0%, *p* = 0.012) and hypocalcemia (12.0% vs. 4.3% or 6.8%, *p* = 0.020) were higher in the vitamin D deficient group than the others. The rates of multi-gestation (13.8% vs. 19.1% or 25.3%, *p* = 0.023) and maternal vitamin D intake (12.9% vs. 23.5% or 34.7%, *p* < 0.001) were lower in the vitamin D deficient group than the others. Maternal vitamin D intake (OR = 0.463, *p* = 0.003), VD (OR = 0.628, *p* = 0.023), and hypocalcemia (OR = 2.409, *p* = 0.033) were associated with neonatal vitamin D deficiency when multivariate logistic regression analysis adjusted for BW was performed (Table 4). The risk of neonatal vitamin D deficiency decreased during SD, but there was no statistical significance (OR = 0.772, *p* = 0.189, Table 4).

### 3.3. Mediation Effect of Social Distancing on Neonatal Morbidity

When we analyzed the impact of SD on neonatal morbidity mediated by neonatal vitamin D status by adjusting for obstetric and perinatal factors, we did not find a significant mediation effect of SD on neonatal morbidity (Table 5).

## 4. Discussion

Unlike changes in vitamin D levels in children and adolescents during the pandemic in previous reports, increased concentrations of serum vitamin D (22.4 ± 11.0 vs. 25.8 ± 12.5 ng/mL, *p* = 0.001) and a decreased rate of vitamin D deficiency in neonates during SD (47.1% vs. 35.4%, *p* = 0.008) were found in this study [16,17,18]. Since neonatal vitamin D status can be affected by maternal vitamin D, an increased rate of maternal vitamin D intake during SD may have contributed to the decreasing rate of neonatal vitamin D deficiency (Table 2 and Table 4) [5,6,23]. Among perinatal or obstetric factors, rates of BF and PPROM decreased but rates of male and maternal vitamin D intake increased during SD (Table 1). A decreasing BF rate during the COVID-19 pandemic due to limited medical services or information regarding the transmission of the virus in breast milk in the early period of this pandemic was reported previously [24,25,26]. Likewise, restriction of visits to prevent infection from parents or breastmilk in our hospital during SD might have contributed to the reduction in the BF rate. An increased rate of male sex was observed during SD compared to B-SD (Table 1 and Table 2). But there was no sex difference according to vitamin D status (Table 3). Male newborn infants have higher morbidity than female infants because of differences in hormonal function and stress responses irrespective of vitamin D deficiency [27]. A marked increase in the rate of maternal vitamin D intake was observed from 2019 (6.8%) to 2021 (37.6%), and SD was a potent related factor in maternal vitamin D intake in this study (Table 2). Among the neonatal factors, the rates of vitamin D deficiency, hypocalcemia, respiratory illness, and neonatal jaundice decreased during SD compared to B-SD (Table 1). However, these factors did not show a significant association with SD when multivariate regression analysis was performed (Table 2).

The rate of neonatal vitamin D deficiency was 41.3% in this study. VD, multi-gestation, maternal vitamin D intake, and SD were significantly different perinatal and obstetric factors between the vitamin D-deficient (25OHD < 20 ng/mL) and -insufficient or -sufficient groups (25OHD ≥ 20 ng/mL, Table 3). Among the significantly different factors according to neonatal vitamin D status, VD and maternal vitamin D intake were significantly associated with neonatal vitamin D deficiency (Table 3 and Table 4). Like previous studies, the results also showed that maternal vitamin D status could affect neonatal vitamin D levels, and maternal supplementation could improve the vitamin D level of the mother and her offspring [5,6,28]. Multiple roles of vitamin D in neonates have been reported besides its fundamental function in bone health, and its deficiency has been associated with many adverse health effects in neonates, including respiratory, infectious, and metabolic diseases [7,8,9,10,11,12,13,14,15]. Although more evidence on routine vitamin D supplementation to pregnant women is necessary, supplementation during pregnancy may help to reduce neonatal vitamin D deficiency without significant side effects [29]. Unlike our results regarding the delivery mode (Table 3), in Danish pregnant women with decreased vitamin D levels, an increased risk of cesarean section (CS) due to dystocia was reported [30]. The decreased rate of VD or increased rate of CS (76.4% vs. 66.8%, *p* = 0.017) in neonates with 25OHD levels of ≥20 ng/mL in this study was not associated with SD (Table 1). In this study, the decreasing rate of VD during SD might be associated with the increased rate of multi-gestation (Table 3). Multi-gestation is one of the high-risk forms of pregnancy, and delivery by CS is often preferred due to higher rates of adverse perinatal outcomes [31]. When we performed a multivariate linear regression analysis for obstetric and perinatal factors associated with the delivery mode, multi-gestation was one of the risk factors for CS (OR = 2.976, 95% confidence interval 1.555–5.694, *p* = 0.001).

Among neonatal morbidities, hypocalcemia was significantly associated with neonatal vitamin D deficiency (Table 3 and Table 4), consistent with previous studies [9,10]. A lower rate of respiratory illness was seen during SD than B-SD (Table 1), but it was not related to neonatal vitamin D deficiency in the study (Table 3), unlike previous reports [11,12,13,14]. When compared to B-SD, the rate of neonatal vitamin D deficiency decreased (47.1% vs. 35.4%, *p* = 0.008) and the mean concentration of 25OHD significantly increased in SD (25.8 ± 12.5 vs. 22.4 ± 11.0 ng/mL, *p* = 0.001), but the level of 25OHD in SD was still lower than 30 ng/mL in this study (Table 1). Therefore, small increases in the mean vitamin D concentrations of neonates during SD, albeit statistically significantly different between B-SD and SD, could not change in the rate of respiratory illness in this study. However, since there were conflicting results regarding the association between vitamin D and respiratory disease in neonates and the mechanism of the effect of vitamin D on respiratory morbidity in neonates remains unclear, the role of vitamin D in RDS, TTN, or BPD needs further investigation [32]. To investigate the mediation effect of SD on neonatal morbidity via neonatal vitamin D status, we used Process Macro model number 4 [22]. However, SD did not significantly affect the change in neonatal morbidity mediated by changes in neonatal vitamin D deficiency (Table 5). In this study, SD imposed by the COVID-19 pandemic might have had an effect on increasing maternal vitamin D intake, which might have also contributed to decreased neonatal vitamin D deficiency (Table 2 and Table 4).

Our study has several limitations. First, this was an observational study conducted in a single hospital. Second, small increases in neonatal vitamin D levels, albeit statistically significant, might have contributed to the insignificant change in neonatal morbidity. Third, the dose or duration of maternal vitamin D supplementation was not investigated. In neonates whose mothers were taking vitamin D (*n* = 117, 28.7 ± 12.2 ng/mL), the mean 25OHD concentration was higher than in the others (*n* = 409, 22.7 ± 11.4 ng/mL, *p* < 0.001). However, even in pregnant women with nutritional vitamin D supplements during pregnancy, the mean 25OHD concentration in their offspring was still insufficient in this study (n = 117, 28.7 ± 12.2 ng/mL). The insufficient vitamin D concentration might be due to the inclusion of hospitalized neonates in this study, which was the fourth limitation of our study. However, this result could indicate that the current commercial dose of maternal vitamin D supplements might be insufficient to prevent neonatal vitamin D insufficiency. Fifth, we did not investigate maternal vitamin D levels and vitamin D-associated lifestyles, such as outdoor activity or the use of sunscreen. Although we could not obtain maternal vitamin D levels directly, the vitamin D level from the offspring’s blood sample within a mean of 0.7 days after birth could reflect the maternal vitamin D condition.

Despite these limitations, our study has several strengths. This was the first report on changes in neonatal vitamin D levels during SD during the COVID-19 pandemic. A decreased rate of vitamin D deficiency was observed, unlike previous reports in children [16,17,18].

## 5. Conclusions

Based on our results, the rate of maternal vitamin D intake significantly increased during SD, which was significantly associated with a decreased rate of neonatal vitamin D deficiency. However, no direct correlation between SD and a decreased rate of neonatal vitamin D deficiency nor a mediating effect of SD on neonatal morbidity was found in this study. Although routine use of micro-nutrients other than folic acid and iron for maternal and fetal health during pregnancy has not been established yet, vitamin D intake may help to increase neonatal vitamin D levels. A commercial dose of vitamin D for pregnant women may not be sufficient to reduce the neonatal morbidity associated with vitamin D deficiency.

## Figures and Tables

**Table 1 nutrients-16-01858-t001:** Clinical characteristics of the subjects according to social distancing.

Clinical Characteristics				Total (*n* = 526)	B-SD (2019, *n* = 263)	SD (2021, *n* = 263)	*p*
Perinatal factors	GA, week		mean ± SD *	36.5 ± 2.7	36.3 ± 2.5	36.7 ± 2.8	0.080
	Birth weight, g		mean ± SD *	2670.3 ± 693.1	2628.5 ± 700.5	2712.1 ± 684.5	0.167
	Male *		*n* (%)	321 (61.0)	143 (54.4)	178 (67.7)	0.002
	VD		*n* (%)	145 (27.6)	67 (25.5)	78 (29.7)	0.329
	Poor AS at 5’		*n* (%)	1 (0.2)	1 (0.4)	0 (0)	1.000
	Breastfeeding *		*n* (%)	128 (25.6)	91 (36.4)	37 (14.8)	<0.001
Obstetric factors	PPROM		*n* (%)	109 (20.7)	65 (24.7)	44 (16.7)	0.031
	Maternal HTN		*n* (%)	78 (14.8)	44 (16.7)	34 (12.9)	0.269
	Maternal DM		*n* (%)	81 (15.4)	42 (16.0)	39 (14.8)	0.809
	Multi-gestation		*n* (%)	98 (18.6)	48 (18.3)	50 (19.0)	0.911
	Maternal vitamin D intake *		*n* (%)	117 (22.2)	18 (6.8)	99 (37.6)	<0.001
	Birth season	Spring	*n* (%)	110 (20.9)	55 (20.9)	55 (20.9)	0.618
		Summer	*n* (%)	134 (25.5)	62 (23.6)	72 (27.4)	
		Fall	*n* (%)	140 (26.6)	69 (26.2)	71 (27.0)	
		Winter	*n* (%)	142 (27.0)	77 (29.3)	65 (24.7)	
Neonatal factors	Vitamin D deficiency *		*n* (%)	217 (41.3)	124 (47.1)	93 (35.4)	0.008
	Hypocalcemia *		*n* (%)	43 (8.2)	33 (12.5)	10 (3.8)	<0.001
	Respiratory illness *		*n* (%)	263 (50.0)	150 (57.0)	113 (43.0)	0.002
	Sepsis		*n* (%)	31 (5.9)	18 (6.9)	13 (4.9)	0.362
	Neonatal jaundice *		*n* (%)	47 (8.9)	31 (11.8)	16 (6.1)	0.033
	NEC		*n* (%)	10 (1.9)	5 (1.9)	5 (1.9)	1.000
	BPD		*n* (%)	7 (1.4)	1 (0.4)	6 (2.3)	0.127
	Mortality		*n* (%)	1 (0.2)	0 (0.0)	1 (0.4)	1.000
	Hospital stays		mean ± SD *	12.9 ± 14.0	13.6 ± 12.8	12.3 ± 15.0	0.310

*p*-values were obtained using the Student’s *t*-test or chi-squared test. * statistically significant factor between the two groups. Abbreviations: B-SD, before social distancing (2019); SD, social distancing (2021); SD *, standard deviation; GA, gestational age; VD, vaginal delivery; poor AS at 5’, 0–3 Apgar score at 5 min after birth; PPROM, prolonged premature rupture of membrane; maternal HTN, maternal hypertension; maternal DM, maternal diabetes mellitus; NEC, necrotizing enterocolitis; BPD, bronchopulmonary dysplasia.

**Table 2 nutrients-16-01858-t002:** Perinatal or neonatal factors associated with social distancing due to COVID-19.

Associated Factors	Odds Ratio	95% Confidence Interval		*p*
		Lower	Upper	
Male	2.008	1.327	3.039	0.001
Breast milk	0.383	0.233	0.629	<0.001
PPROM	0.573	0.348	0.941	0.028
Maternal vitamin D intake	8.735	4.854	15.719	<0.001
Neonatal Vitamin D deficiency	0.751	0.497	1.135	0.174
Hypocalcemia	0.459	0.206	1.023	0.057
Neonatal jaundice	0.482	0.219	1.062	0.070
Birth weight	1.00	1.00	1.001	0.206

*p*-values were obtained using stepwise backward multivariate linear or non-linear regression analysis adjusted for birth weight. Abbreviations: COVID-19, coronavirus disease 2019; PROM, prolonged premature rupture of membrane.

**Table 3 nutrients-16-01858-t003:** Clinical characteristics of subjects according to neonatal vitamin D deficiency.

Clinical Characteristics				Total (*n* = 526)	25OHD (ng/mL)			*p*
					<20 (*n* = 217)	20–30 (*n* = 162)	≥30 (*n* = 147)	
Perinatal factors	GA, week		mean ± SD	36.5 ± 2.7	36.6 ± 2.6	36.6 ± 2.7	36.2± 2.7	0.367
	Birth weight, g		mean ± SD	2670.3 ± 693.1	2727.8 ± 680.5	2676.5 ± 707.3	2578.5 ± 691.0	0.130
	Male		*n* (%)	321 (61.0)	138 (63.6)	99 (61.1)	84 (57.1)	0.464
	VD		*n* (%)	145 (27.6)	72 (33.2)	45 (27.8)	28 (19.0)	0.012
	Poor AS at 5’		*n* (%)	1 (0.2)	0 (0.0)	1 (0.6)	0 (0.0)	0.586
	Breast milk		*n* (%)	128 (25.6)	57 (27.7)	44 (28.4)	27 (19.4)	0.145
Obstetric factors	PPROM		*n* (%)	109 (20.7)	48 (22.1)	35 (21.6)	26 (17.7)	0.586
	Maternal HTN		*n* (%)	78 (14.8)	32 (14.7)	23 (14.2)	23 (15.6)	0.927
	Maternal DM		*n* (%)	81 (15.4)	35 (16.1)	27 (16.7)	18 (12.9)	0.628
	Multi-gestation		*n* (%)	98 (18.6)	30 (13.8)	31 (19.1)	37 (25.2)	0.023
	Maternal vitamin D intake		*n* (%)	117 (22.2)	28 (12.9)	38 (23.5)	51 (34.7)	<0.001
	Birth season	Spring	*n* (%)	110 (20.9)	42 (19.4)	34 (21.0)	34 (23.1)	0.740
		Summer	*n* (%)	134 (25.5)	56 (25.8)	39 (24.1)	39 (36.5)	
		Fall	*n* (%)	140 (26.6)	53 (24.4)	46 (28.4)	41 (27.9)	
		Winter	*n* (%)	142 (27.0)	66 (30.4)	43 (26.5)	33 (22.4)	
	Social distancing		*n* (%)	263 (50.0)	93 (42.9)	80 (49.4)	90 (61.2)	0.003
Neonatal factors	Hypocalcemia		*n* (%)	43 (8.2)	26 (12.0)	7 (4.3)	10 (6.8)	0.020
	Respiratory illness		*n* (%)	263 (50.0)	115 (53.0)	77 (47.5)	71 (48.3)	0.508
	Sepsis		*n* (%)	31 (5.9)	18 (8.3)	6 (3.7)	7 (4.8)	0.136
	Neonatal jaundice		*n* (%)	47 (8.9)	13 (6.0)	16 (9.9)	18 (12.2)	0.109
	Seizure		*n* (%)	8 (1.5)	4 (1.8)	3 (1.9)	1 (0.7)	0.674
	NEC		*n* (%)	10 (1.9)	4 (1.8)	1 (0.6)	5 (3.4)	0.212
	BPD		*n* (%)	7 (1.4)	3 (1.4)	1 (0.6)	3 (2.1)	0.585
	Mortality		*n* (%)	1 (0.2)	0 (0.0)	1 (0.6)	1 (0.7)	0.390
	Hospital stays		mean ± SD	12.9 ± 14.0	12.9 ± 14.0	12.8 ± 13.1	13.0 ± 15.0	0.962

*p*-values were obtained using the one-way analysis of variance or chi-squared test. Abbreviations: 25OHD, 25-hydroxy vitamin D; SD, standard deviation; VD, vaginal delivery; poor AS at 5’, 0–3 of Apgar score at 5 min after birth; PPROM, prolonged premature rupture of membrane; maternal HTN, maternal hypertension; maternal DM, maternal diabetes mellitus; NEC, necrotizing enterocolitis; BPD, bronchopulmonary dysplasia.

**Table 4 nutrients-16-01858-t004:** Perinatal or neonatal clinical factors associated with neonatal vitamin D deficiency.

Associated Factors	Odds Ratio	95% Confidence Interval		*p*
		Lower	Upper	
Vaginal delivery	0.628	0.420	0.938	0.023
Multi-gestation	1.327	0.782	2.253	0.295
Maternal vitamin D intake	0.463	0.280	0.769	0.003
Social distancing	0.772	0.525	1.136	0.189
Hypocalcemia	2.409	1.060	3.960	0.033
Birth weight	1.000	1.000	1.000	0.277

*p*-values were obtained using stepwise backward multivariate regression analysis adjusted for birth weight.

**Table 5 nutrients-16-01858-t005:** Mediation effect of social distancing on neonatal morbidity mediated by neonatal vitamin D deficiency.

Neonatal Morbidity	Estimate	95% Confidence Interval		*p*
		Lower	Upper	
Respiratory illness	0.002	−0.016	0.002	0.856
Sepsis	−0.043	−0.099	0.013	0.129
Neonatal jaundice	0.036	−0.014	0.086	0.158
Hypocalcemia	−0.047	−0.102	0.009	0.100
BPD, mod to severe	0.000	−0.119	0.119	0.998

*p*-values were obtained from Process Macro model number 4. Abbreviations: BPD, bronchopulmonary dysplasia.

## Data Availability

Data are contained within the article and Appendix A

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
