# Peer review of "Mediation Effect of Social Distancing on Neonatal Vitamin D Status and Related Clinical Outcomes during the Coronavirus Disease-19 Pandemic"

_nutrients, 2024, doi:10.3390/nu16121858_

Round 1

Reviewer 1 Report

Comments and Suggestions for Authors

In this study Jun et al conducted an interesting retrospective study on analyzing the impact of social distancing (SD) on vitamin D status and the associated morbidity in neonates during the coronavirus disease (COVID)-19 pandemic.

Please see below some comments to help you with the improvement of the manuscript.

General comments:

Lines 77-79: shuld be presented in results section.

Lines 91-94: are a statement after data analysis. How did the authors reach this conclusion?????

What was the dose of vitamin D that the mothers received? was there any difference? are there data on maternal vitamin d levels? What was the dose of vitamin D that the mothers received? was there any difference? are there data on maternal vitamin d levels? I think this information is important enough to be evaluated the impact of the impact of social distancing (SD) on vitamin D status and the associated morbidity in neonates during the coronavirus disease (COVID)-19 pandemic.

Data in table 3 shuld be presented in diferent way : 25OHD (ng/mL) < 20; 20 -30; and >30 and in assosiation to time periods.

Author Response

Thank you for taking the time to review our manuscript and provide valuable feedback. We carefully considered the reviewer’s comments and tried to incorporate the suggestions. Our point-by-point responses are presented below.

1. Lines 77-79: shuld be presented in results section.

  • We moved the sentence from a section of materials and methods to that of results.

2. Lines 91-94: are a statement after data analysis. How did the authors reach this conclusion?????

  • The definitions of RDS and TTN were referenced from “Nelson Textbook of Pediatrics, 21th edition”. And we cited the chapter of the textbook according to the reviewer’s comment. [Ahlfeld SK. Respiratory Tract Disorder. In: Kliegman RM, Blum NJ, Shah SS, St. Geme JW, Tasker RC, Wilson KM, Behrman RE, eds. Nelson Textbook of Pediatrics. 21th ed. Philadelphia: Elsevier, 2020:929-949].
  • We reviewed the previous reports regarding the association of vitamin D and respiratory disease in neonates. RDS often occurs in preterm infants and TTN in late preterm or term infants. The both respiratory diseases associated with vitamin D deficiency were reported previously [references 11-14]. And we enrolled both preterm and term infants and there were no statistical differences in the rates of preterm infants between the two periods (before SD and during SD). Based on the references regarding the association of vitamin D and RDS or TTN in neonates which we cited [11-14] and no statistical difference in the rate of preterm infants (preterm infants born at less than 32 weeks of gestation; 8% in B-SD vs. 8.4% in SD, P = 0.622), we considered RDS and TTN as one factor of respiratory illness.

3. What was the dose of vitamin D that the mothers received? was there any difference?

The reviewer’s comment is one of the limitations in the study. We could not obtain the detailed dose of vitamin D that the mothers received. However, babies of mothers took two brands of vitamin supplements (n=10) had higher vitamin D levels than single brand supplement or no supplement (Kruskal-Wallis test). Because of the small number of women who took two vitamin brands, we did not consider another factor to be analyzed. 

No. of maternal vitamin D supp

N

Baby's viamin D level Mean

Std. Deviation

Std. Error

P

0

409

22.7359

11.43308

0.56533

<0.001

1

107

27.8206

12.32691

1.19169

2

10

38.4600

4.57292

1.44608

Total

526

24.0692

11.86678

0.51742

4. are there data on maternal vitamin d levels? What was the dose of vitamin D that the mothers received? was there any difference? are there data on maternal vitamin d levels? I think this information is important enough to be evaluated the impact of the impact of social distancing (SD) on vitamin D status and the associated morbidity in neonates during the coronavirus disease (COVID)-19 pandemic.

  • Thank you for the comments. We added the reviewer’s comment in the limitations of the study as follows: Fifth, we did not investigate the maternal vitamin D level and maternal vitamin D-associated lifestyle factors, such as outdoor activity or the use of sunscreen. Although we could not obtain maternal vitamin D levels directly, the vitamin D level from the offspring’s blood sample within a mean of 0.7 days after birth could reflect the maternal vitamin D condition.

5. Data in table 3 shuld be presented in diferent way : 25OHD (ng/mL) < 20; 20 -30; and >30 and in assosiation to time periods.

  • Thank you for the recommendation. We analyzed the clinical characteristics of inclusions according to vitamin D deficiency (<20), insufficiency (20-30) and sufficiency (>= 30 ng/mL) and the results were presented in Table 3. And the materials and methods, results and discussion in the manuscript were updated according to the comparisons in Table 3.

Reviewer 2 Report

Comments and Suggestions for Authors

Thank you for giving me the opportunity for review the manuscript entitled ” Mediation Effect of Social Distancing on Neonatal Vitamin D Status and Associated Clinical Outcomes during the Coronavirus Disease-19 Pandemic”

The manuscript fits within the scope of the journal but requires some clarifications.

The topic is of interest, as pregnant women have special  requirements that need to be met to prevent damage to the growing fetus as well as to prevent the development of certain diseases later in live.

Please find the specific comments below:

In the discussion section, the authors point out a number of important limitations that significantly affect the quality of the work.

1.       Is it not possible to obtain information from medical records about supplements taken by the mother during pregnancy?

2.       Should the place of residence (rural, urban) or the duration of DS not be included in the analyses?

3.       What are the recommendations for the use of dietary supplements in South Korea?

Is it possible to add this information?

Author Response

<Responses to Reviewer 2>

Thank you for taking the time to review our manuscript and provide valuable feedback. We have carefully considered the reviewer’s comments and tried our best to incorporate the suggestions. Our point-by-point responses are presented below.

Please find the specific comments below:

In the discussion section, the authors point out a number of important limitations that significantly affect the quality of the work.

  1. Is it not possible to obtain information from medical records about supplements taken by the mother during pregnancy?

-> Because vitamin supplementation have not yet been covered by national medical insurance in South Korea, there is no information of prescribing vitamin D in maternal medical records. Therefore, we could obtain the information on taking vitamin D by asking whether the mother had been taking vitamin D during pregnancy when the offspring was admitted to our NICU.

  1. Should the place of residence (rural, urban) or the duration of DS not be included in the analyses?

-> Our hospital is surrounded by small and medium-sized town and we did not consider the residence type.

-> What is the meaning of duration of DS that the reviewer mentioned?

If it means duration of social distancing in South Korea, it was over two years from March 2020 to April 2022 which was explained in the section of introduction.

If it means duration of maternal vitamin D supplementation, we could not obtain the information of the duration which was the one of the limitations of the study and described in the section of discussion.

  1. What are the recommendations for the use of dietary supplements in South Korea?

Is it possible to add this information?

  • The national medical insurance of South Korea approved the supplementation of iron and folic acid for maternal and fetal health during pregnancy. We added the information in the section of conclusion as follows. Although routine use of micro-nutrients other than folic acid and iron for maternal and fetal health during pregnancy has not been established yet, vitamin D intake may help to increase neonatal vitamin D levels.
